# Associations between Parental Stress and Subsequent Changes in Dietary Intake and Quality among Preschool Children Susceptible to Obesity

**DOI:** 10.3390/ijerph18073590

**Published:** 2021-03-30

**Authors:** Jeanett Friis Rohde, Sofus Christian Larsen, Mina Nicole Händel, Nanna Julie Olsen, Maria Stougaard, Berit Lilienthal Heitmann

**Affiliations:** 1Research Unit for Dietary Studies at the Parker Institute, Bispebjerg and Frederiksberg Hospital, 2000 Frederiksberg, Denmark; sofus.larsen@regionh.dk (S.C.L.); Mina.Nicole.Holmgaard.Handel@regionh.dk (M.N.H.); nanna.julie.olsen@regionh.dk (N.J.O.); Berit.Lilienthal.Heitmann@regionh.dk (B.L.H.); 2Center for Early Interventions and Family Studies, Department of Psychology, University of Copenhagen, 1353 Copenhagen K, Denmark; maria.stougaard@psy.ku.dk; 3The Boden Institute of Obesity, Nutrition, Exercise & Eating Disorders, The University of Sydney, Sydney 2006, Australia; 4Department of Public Health, University of Copenhagen, 1014 Copenhagen K, Denmark

**Keywords:** family stress, children, dietary intake, diet quality

## Abstract

Background: Cross-sectional studies indicate that parental stress may be a barrier for healthy dietary behaviours among children. However, there is a lack of evidence from longitudinal studies on the association between parental stress and changes in dietary intake among toddlers. The aim of this study was to examine the association between parental stress and changes in dietary intake and quality among preschool children susceptible to obesity. Methods: In the Healthy Start study, parents to 250 preschool children had completed a modified version of the Parental Stress Index and assessed the dietary intake of their children at baseline and after 15 months of follow up. The association between parental stress and changes in dietary intake and quality was examined using multiple linear regression analyses with adjustment for potential confounders. We tested for potential effect modification by group allocation and sex. Results: There were no significant associations between parental stress and subsequent changes in child total energy intake, intake of macronutrients or intake of fruit, vegetables, sugar sweetened beverages, fish or starch, or dietary quality. Conclusion: This study provides no evidence to support an association between parental stress and subsequent change in dietary intake and quality of their children. Trial registration: ClinicalTrials.gov, Trial number: NCT01583335, Registered: 31 March 2012, retrospectively registered.

## 1. Introduction

The past decades have been characterized by an increasing global obesity epidemic, especially among young children with around 41 million children under the age of five having overweight [1]. It is known that dietary and physical activity behaviors are important contributing factors to the obesity epidemic and thus research has extended its view beyond this traditional concept and are now also investigating other factors such as stress. Previous studies have found a relationship between stressful events or situations such as social isolation, parental neglect, lack of support from family or other social contacts, natural disasters or major life-events and development of overweight and obesity in childhood, which potentially can be explained by changes in dietary patterns towards more unhealthy choices [2,3,4,5]. An explanation could be that children may use snacking as a stress coping mechanism [6,7]. The hypothesis is that during periods with interpersonal stress some children may decrease their food intake while others, especially the high dietary restrained children, may increase intake of energy dense food, which may lead to obesity [8].

Parents have a high degree of influence on their child’s dietary behavior [9,10], and it is argued that also parental stress may have a spillover effect on the health behavior of their offspring, especially in relation to obesity risk [11,12]. Parental stress may indirectly influence child dietary intake through unhealthy parental behavior due to responses to stressors. Parents who are stressed may have a higher intake of unhealthy food and engage less in physical activity leading to increased sedentary behavior [13]. Furthermore, parents who are stressed may have less time to prepare healthy meals leading to unhealthy eating practices, and thereby indirectly influence child risk of obesity through modeling of unhealthy dietary behaviors [13,14]. In a cross-sectional study by Parks et al. [4], a direct association was reported between perceived parental stress and a higher fast-food consumption among their children at ages 3–17 years. This was further supported by a cross-sectional study from Bauer et al. that reported that parental employment and workload contributed to decreased time to prepared meals and less frequent family meals as well as more unhealthy food choices [13]. Another cross-sectional study found that among the children, higher hair cortisol concentrations were associated with a lower consumption of dietary fat, however no association was observed between parental hair cortisol concentrations and dietary intake and quality among their children [15]. Hence, both child and parental stress level may be barriers for healthy dietary behaviours among children, which are necessary for long term weight balance. 

Despite some evidence from cross-sectional studies showing an association between parental stress and child dietary intake, knowledge from longitudinal studies on the association between these underlying factors contributing to the obesity epidemic is sparse among preschool children.

To address these gaps, the current study used longitudinal data from the “Healthy Start” study [16] to examine the association between parental stress and subsequent changes in dietary intake and quality among preschool children predisposed to obesity. 

## 2. Materials and Methods

The present study derived from the Danish Healthy Start primary intervention study [16] and the results presented stem from the prospective data collected over time. The “Healthy Start” investigated the joint effects of increasing physical activity, improving dietary habits, improve sleep habits and manage stress on child obesity prevention (ClinicalTrials.gov; NCT01583335). The intervention took place over on average period of 15 months and was conducted between 2009 and 2011. Families in the intervention group were offered the possibility to have up to ten sessions with a health consultant over a 15-month period focusing on stress, sleep, dietary intake and physical activity behaviour. The intervention was tailormade, meaning that the consultations were based on the family’s specific needs and resources. Families were also invited to participate in bi-monthly cooking and monthly play events, which took place at local schools in five of the participating municipalities. Families in the control group were invited to participate in a baseline examination and invited to follow-up examination approximate 15 month later and were not examined in between.

Participants lived in 11 municipalities around the greater Copenhagen area and were between 2–6 years old at study entry. Eligibility criteria were being healthy weight but susceptible to future overweight defined as (i) having a high birth weight (>4000 g) and/or (ii) having a mother who was overweight prior to pregnancy (body mass index (BMI) ≥ 28 kg/m^2^) [17]. A subgroup from one municipality was selected based on the criterion of having a mother with low educational level (≤10 years of education) [17,18]. 

### 2.1. Study Population

The “Healthy Start” intervention study initially invited 3,722 children aged 2–6 years to participate in the intervention or control group, of which 21% of the parents agreed to participate [16]. Data in the present study is from the baseline and follow-up examinations of 635 children, 320 in the intervention group and 315 children in the control group. For the present study, children with missing information on the exposure variable (parental stress), the outcome variables (dietary intake) or any of the potential confounders (group allocation, age, sex, body mass index (BMI), sleep duration, physical activity, maternal factors; education, BMI and physical activity level) were excluded, and therefore the final study population for analysis consisted of 250 (n = 112 intervention group; n = 138 control group) children aged 2–6 years (Figure 1). 

### 2.2. Measures

#### 2.2.1. Parental Stress

Information on parental stress was obtained through a questionnaire completed by one of the parents at baseline of the “Healthy Start” study [16]. The Parental Stress Index, which is a self-reported inventory, designed to measure parental experiences of stress in relation to the parent–child interaction was used to assess parental stress. In the Healthy Start project, a modified Swedish version [19] was included, as an elaboration of the parental part of the questionnaire. In total 10 questions were selected from the Swedish version and modified according to context (Appendix A).

The questions concerned which changes the parents had experienced in their daily living after they had had the child, in relation to sleep, stress, worries, time for themselves, household conflicts, workload, social gatherings in the home, joy of life, energy in daily living, and perceived complexity of being a parent compared to expectations. 

Answers were coded 0–2 with 0 being the best score and 2 the worst score, e.g., less stress = 0, more stress = 2 or no change in stress level = 1. Analyses of intercorrelations and factor analyses showed that nine (workload excluded) out of the ten questions could be added together to an overall score measure of parental stress [20], with a higher score indicating higher levels of perceived stress among the parents. 

#### 2.2.2. Dietary Intake

Children’s dietary intake was assessed using a four-day estimated food diary (Wednesday to Saturday) completed by the parents at baseline and after the 15 months of follow-up. These specific days were chosen to obtain information on dietary intake in both week- and weekend days. For each day the parents should register which food item was consumed, portion size, product brand and if possible, the fat content of the food item or meal. Furthermore, the parents should register the place and time for where and when the meal or food item was consumed. To help the parents estimate portion sizes, the food diaries were accompanied by a picture book including seventeen photo series with foods and portion sizes [21]. All information on dietary intake was manually entered, by one of the investigators, to the software program Dankost 3000 (http://dankost.dk, accessed on 4 February 2021, Dansk Catering Center Denmark) and nutritional calculations were performed to calculate the daily intake of macronutrients in grams and energy percent (E%). This software is based on the official Danish national food composition database (version 7.01) developed by the National Food Institute at the Technical University of Denmark [22]. Afterwards, means of total energy and macronutrient intake based on the four-day recorded food diaries was estimated for each child. Dietary intake of total energy, protein, fat, saturated fat, carbohydrate and added sugar was included in the analyses in units of KJ/day or as E%. 

As it was not possible to define food groups using the software Dankost 3000, this was done manually by studying a list of all food items eaten by the children. An algorithm was developed and programmed in Stata (Statacorp corporation, College Station, TX, USA) to be able to extract information and generate food groups (fruit, vegetables, sugar sweetened beverages (SSB), fish and starch). Food groups were defined based on standards developed by the National Food Institute, Technical University of Denmark [22] and adjusted to fit the Danish dietary guidelines (Appendix A). Dietary intakes of fruit, vegetables, SSB, fish and starch were included in the analyses in units of g/day. Moreover, a diet quality index (DQI), adapted from Knudsen and colleagues [23], was calculated to examine to what degree the children complied with the Danish national dietary guidelines [24]. The DQI score was calculated based on six nutrients/food groups, as a function of recommended vs. reported intake, and included the following nutrients and food items: fat (maximum 30 E%), saturated fat (maximum 10 E%), added sugar (maximum 10 E%), fish (minimum 200 g/week), fruit and vegetables (minimum 300 g/day) and potatoes, rice or pasta (minimum 200 g/day). For the food groups with a minimum recommended intake, the group-score would be based on the ratio (R/R_T_) between the reported (R) and the recommended (R_T_) intake; where the score was set to 1 for intakes with R ≥ R_T_. For the food groups with a maximum recommended intake, the score was instead derived as 1 − (R − R_T_)/(R_max_ − R_T_); where the score is set to 1 for intakes R ≤ R_T_. The total DQI score ranged from 1 to 6, with 1 referring to values furthest away from following the recommendations, and with six referring to best compliance with the Danish national dietary recommendations. 

#### 2.2.3. Confounders

The key confounders were selected a priori based on information from the existing literature and comprised group allocation (intervention/control status), child age, sex, BMI Z-score, sleep duration, physical activity as well as maternal factors such as; education, BMI and physical activity level. Information on the child’s gender and age, was obtained from the Danish Medical Birth Registry [25,26]. Information on sleep (nabs not included) and physical activity patterns of the child derived from the baseline questionnaire completed by one of the parents. Parents were asked to report the time that their child fell asleep in the evening and woke up in the morning from Monday to Sunday to capture sleep duration on both weekdays and weekend days. Afterwards, average nighttime sleep duration was calculated as the mean in hours of all registered days at baseline. Physical activity level were defined based on a single question: “*How physically active is the child compared to other children at the same age*”. The parents could answer if they perceived their child as being “*not so active*”, “*fairly active*”, “*very active*” or “*Don’t know*”. For this study, information was collapsed into two categories ”*less active*” and “*very active*” and those answering “Don’t know” whereas omitted (n = 8). Children’s height and weight were measured by trained health professionals at baseline, using a mechanical weight or beam-scale type weight (TanitaBWB-800, Tania, Amsterdam, The Netherlands) or SV-SECA 710 (SECA, Hamborg, Germany) and stadiometers (Soehnle 5002, Soehble, Backnang, Germany) or Charter ch200P and were used to calculate child BMI. From that measure BMI z-scores were generated using the Lambda-Mu-Sigma method, which summarizes the changing distributions of the dependent variable (e.g., BMI) by the median, the coefficient of variation and skew expressed as Box-Cox power [27]. A power transformation in increments of 0.1 years was used, applying national reference BMI Z-scores to the study population [28].

Maternal BMI was based on self-reported height and weight and was calculated by dividing weight in kg with height in meters squared (kg/m^2^). Mothers also reported on their highest education level in nine categories which for the study were collapsed into four categories: (1) low education level (“primary and lower secondary”, “upper secondary”, “one or more short courses” and “skilled worker”), (2) medium education level (“short-term further education [<3 years]” or “medium-term further education [3–4 years]”), (3) High education level (“long-term further education [>4 years]”, “research level”) and (4) other education (educational information which was not possible to classify according to above). Maternal physical activity level was also self-reported and defined based on a single question “*Which of the following categories best describes your level of physical activity in your spare time, seen over the last year*”. The parents could answer if they (1) conducted vigorous physical activity several times per week, (2) did moderate physical active at least 4 h per week (exercise, bike to work, heavy gardening), (3) light physical activity at least 4 h per week (walking, light gardening) or (4) primarily sedentary activity. For this study, information was collapsed into two categories “*trained hard serval times per week*” (answer 1 and 2) and “*being physical active for at least 4 times per week*” (answer 3 and 4).

### 2.3. Ethical Approval and Consent to Participate 

Written informed consent to use the collected data for research purpose was obtained from all the participants’ parents or legal guardians (Appendix A). The study complied with the Helsinki II declaration and The Danish Data Protection Agency approved of the study (journal number: 2015-41-3937). The Scientific Ethical Committee of the Capital Region in Denmark decided that according to the Danish law, the study was defined not to be a bioethics project and did not need further approval (journal number H-A-2007-0019) [16]. 

### 2.4. Statistical Methods

Descriptive analyses were carried out on baseline and follow-up measures of exposure and outcome and on baseline measures of confounders. Multiple linear regression models were used to assess the associations between parental stress and subsequent changes in dietary intake and dietary quality index with adjustments for baseline measure of outcome (dietary intake or quality), group allocation (intervention/control), age, sex, BMI Z-scores, sleep duration, physical activity level and maternal education, BMI and physical activity.

Potential effect modification by group allocation (intervention/control group) and sex was explored for all outcomes by adding a product term (baseline outcome x group allocation or sex) to the models. If interactions were significant (*p* < 0.05), further evaluation was performed through stratified analyses.

Sensitivity analyses: Especially at follow-up, we had a relatively large proportion of participants with missing information. Thus, in sensitivity analyses multiple imputations were used to impute missing values for all participants who had complete information on exposure and outcomes at baseline (n = 495). The imputations were made using chained equations as implemented in Stata through the commands ice and mim [29]. For the multiple imputations, m = 10 complete datasets were first generated. Baseline measure of dietary intake in addition to all preselected bassline covariates were used as imputation variables. In each dataset, the missing values was replaced with the imputed values, which were constructed based on predictive distributions for each of the missing values. Afterwards each of the completed datasets was analysed, and the results from the ten analyses were combined to create a single set of estimates comprising the variability associated with the missing values. As in the primary analyses, linear regressions were used to explore both crude and adjusted associations.

All statistical analyses were two-sided with a significance level at 0.05 and were performed using Stata SE 14 (StataCorp LP, College Station, TX, USA; www.stata.com, accessed on 4 February 2021).

## 3. Results

The participants’ demographics are outlined in Table 1, presented as edian and 5–95% percentiles unless stated otherwise. Slightly more boys (56%) than girls participated, median age was 4.1 years (5–95% percentiles: 2.4; 5.7) and had a median BMI Z-score of 0. (5–95% percentiles: −0.9; 1.7). The median parental stress score was 14 at baseline with a range from 10–18 points. The children’s total energy intake and intake of macronutrients followed the national Danish dietary recommendations [24], except for saturated fat intake which were above the recommendation of max 10E% per day (Median: 11.2 E%, 5–95% percentiles: 7.0; 14.7 E%).

Table 2 presents the association between parental stress and subsequent changes in child dietary intake and dietary quality index. We found no evidence for an association between parental stress and subsequent changes in total energy intake (Adjusted β −17.05 (95% CI: −56.45; 53.06)). Furthermore, there was no association between parental stress and changes in intake of any of the macronutrients (Table 2). 

Moreover, no associations between parental stress and changes in fruit intake (Adjusted β 0.43 (95% CI: −2.56; 3.43)), vegetables intake (Adjusted β −2.07 (95% CI: −5.38; 1.25)), fish intake (Adjusted β 0.64 (95% CI: -0.40; 1.67)), starchy foods intake (Adjusted β −1.16 (95% CI: −2.97; 0.64)) or SSB intake (Adjusted β 0.99 (95% CI: −3.75; 5.73)) were observed in either the crude or adjusted analyses (Table 2).

Additionally, the association between parental stress and subsequent changes in dietary quality index, were not significant (Adjusted β 0.00 (95% CI: −0.02; 0.03)) (Table 2). The results were essentially similar in the sensitivity analyses where data were imputed on those participants who had complete information on exposure and outcomes at baseline (Appendix A).

We further found no evidence of interaction between group (intervention/control) or child gender and parental stress in relation to intake of energy, macronutrients, food groups or DQI. 

## 4. Discussion

The present longitudinal study examined the association between parental stress and changes in child dietary behavior over 15 months among preschool children susceptible to development of overweight. Surprisingly, our findings suggest no evidence of an association between parental stress and subsequent changes in child dietary behaviors or dietary quality index. These results are somewhat in contrast with a previous longitudinal study, including 117 mother-offspring pair [30], in which they reported that among preschool children, there was an inverse association between high perceived maternal parenting stress, measured by Parent–Child Dysfunctional Interaction subscale of the Short Form of the Parenting Stress Index at baseline, and mother-reported decreased intake of vegetables in the child. However, similar to our findings, there were no associations found with changes in offspring intake of fruit or unhealthy food habits over one year [30]. Thus, more research from longitudinal studies is needed to confirm these findings.

Our results are also in contrast to the cross-sectional evidence, that for the most part indicated that a higher level of parental perceived stress was associated with suboptimal diet among the offspring [31]. Specifically, previous cross sectional studies have showed conflicting results ranging from no association to associations of higher parental perceived stress with higher intake of unhealthy food, such as child fast food intake, and in some cross-sectional studies, but not all, a lower intake of healthy food such as fruit and vegetables among the offspring [4,31]. Another cross-sectional study [32] found that the more overall stress the mother perceived, measured by three subscales of the Trierer Inventory of Chronic Stress tool, the less core foods (muesli, salad, fruits, vegetables, dairy products, and fish) and the more non-core foods (chocolate, sweets, cakes, and salty snacks) the child consumed. Hurley et al. 2015 [33] used the Perceived Stress Scale to measure maternal stress and found a direct association between a high perceived stress level and earlier introduction of solid foods (before 4 months of age) and with adding cereal to the infant’s bottle [33]. However, no associations were found between maternal stress and nutrient intakes for infants aged 7–12 months [33]. 

Hair cortisol concentration is a relatively new biomarker [34,35], providing a biological measure of long-term cortisol exposure as cortisol is embedded into hair as it grows and when comparing to objective measures of parental stress, the results from this current study are in accordance with our results investigating the cross-sectional association between parental hair cortisol levels and intake of total energy, added sugar, selected food groups or DQI among their children, where we also could not demonstrate and association among 296 children [15].

In summary it seems likely that the different conclusions from different studies, can be explained by the different study designs and definitions of both exposures and outcomes. Indeed, there are a wide variety of inventories to measure parental stress measurements, which illustrates the diverse criteria used to operationalize stress. Studies, for example, measured stress by the Parenting Stress Index, the Perceived Stress Scale, the Trierer Inventory for Chronic Stress [31] or as we did, a modified version or a subscale focusing of a specific domains for example financial strain, mental health or stress within the family [31]. The same variation was seen for measurements of child dietary intake where several different instruments were used to assess child dietary intake. The most commonly used instruments were validated food frequency questionnaires (FFQ), the Children’s Eating and Physical Activity Questionnaire or dietary recalls, where dietary intake is recorded retrospective [31]. Other studies used targeted questionnaires to assess children’s intake of specific food items such as fruit, vegetables, fish, fast food or sugar sweetened beverages or categorized food items as either unhealthy or healthy. Currently, there is no consensus regarding the best method to assess dietary intake in young children and all method are influenced by information bias leading to either over- or underreporting of dietary intake [36,37], although some research indicate that among children below the age of five, weighed food records provide the best estimate for total energy intake, when compared to doubly labelled water [37]. The disadvantages by using this method are, however, that it requires more time and engagement from the parents than other methods and therefore may influence compliance. Opposite, FFQs have been designed to capture usual food intake and is considered good at capturing long-term changes in dietary patterns such as intake of specific food items or groups and may be more useful for larger studies because of a relatively lower participant burden [31]. The disadvantage is that the FFQs are less detailed in regard to nutrients intake and require participants to remember and generalise past food intake [38]. Furthermore, FFQ’s have a tendency to generally give higher estimates of dietary intake compared to food records [36]. Therefore, when assessing dietary behaviours among children, it has been suggested that researchers firstly consider the participant burden and secondly gains in accuracy and precision [38]. 

A central strength of the current study includes the access to longitudinal data, thus limiting the potential influence of reverse causality. Furthermore, the study included detailed information on dietary patterns together with information on other lifestyle factors about the family, allowing to comprehensively adjust for potential confounders.

There are also some limitations, including a small sample size and the use of parent-reported information for child dietary intake, which may have contributed to attenuating our associations. Methodological limitations, such as lack of power may have contributed to the lack of observed associations. Furthermore, we did not have sufficient statistical power to stratify the analyses according to selection criteria or to investigate differences between those with extreme high scores of parental stress compared to those with extreme low parental stress scores. Finally, we did not have the necessary statistical power to investigate individual maternal and paternal effects. Future studies with larger samples are therefore warranted to get a better understanding of the possible associations. Moreover, selection bias may also be a concern because of missing data due to attrition, as we only had complete information on 39.4% of the cohort. However, analyses of non-completers, comparing children that participated in the Healthy Start study with those not participating, showed no differences between completes and non-completers with respect to child sex, BMI, levels of PA, dietary intake and in relation to maternal age, BMI, education and educational level. Though, a lower mean age and lower intake of SSB was observed among completers compared to non-completers [39].

For this study we used a subjective measure of perceived parental stress, thereby relying on self-reported information, which may have led to information bias in the form of misclassification of stress level due to limited self-insights. This potential misclassification introduced using subjective evaluated information may potentially have led to attenuation of our observed associations.

Moreover, the questionnaire to measure parental stress was modified from the Swedish version of the Parental Stress Index parental and hence was not validated, which may lead to potential measurement error and may have further attenuated our associations. Furthermore, we had no information on which parent (mother or father) completed the questions on perceived stress. However, we suspect that the ranking of stress level among mothers and fathers to be the same and by that not affecting our results.

Caution should be taken when generalizing the results to all pre-school children because our study population was composed of children all susceptible to develop overweight later in life. Moreover, our results may not be genialized to children from less affluent societies as the participating families were from a society with generally high affluence. 

## 5. Conclusions

In conclusion our longitudinal study provided no evidence to support an association between parental stress and a subsequent change in child dietary intake and quality. These findings imply that parental stress may not substantially influence dietary intake in young children. However, more research from other longitudinal studies is needed. Furthermore, studies with more comparable measures of stressors and dietary intake are recommended. 

## Figures and Tables

**Figure 1 ijerph-18-03590-f001:**
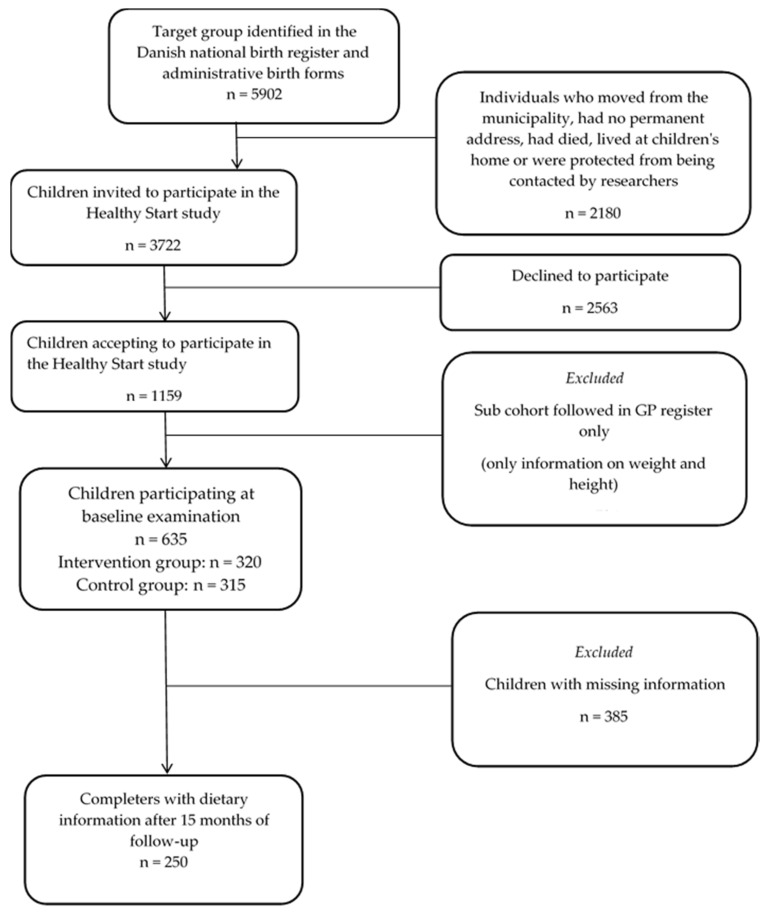
Flowchart of the study population.

**Table 1 ijerph-18-03590-t001:** Study characteristics at baseline and after 15-month follow-up. Results presented as median and 5–95 percentiles unless stated otherwise.

Characteristics	Baseline	Follow-Up
N	250	250
Parental stress (points)	14 (10; 18)	16 (12; 19)
Age (years)	4.1 (2.4; 5.7)	5.4 (3.6; 7)
Gender (% boys)	56	56
Total energy (MJ/day)	4.9 (3.4; 6.8)	5.4 (3.8; 7.5)
Protein (E%)	15.4 (12.4; 20.1)	15.5 (11.9; 20.4)
Carbohydrates (E%)	54.9 (46.6; 62.9)	53.4 (45.3; 62.4)
Added sugar (E%)	7.0 (1.4; 13.3)	6.4 (2.2; 14.0)
Fat (E%)	29.7 (22.4; 38.3)	30.4 (22.2; 38.9)
Saturated fat (E%)	11.2 (7.0; 14.7)	11.2 (7.0; 14.7)
Fruit (g/day)	91.3 (22.6; 212.7)	89.4 (9.8; 228.9)
Vegetables (g/day)	94.4 (21.6; 192.4)	102.2 (25.0; 241.5)
Fish (g/day)	12.1 (0.0; 49.8)	11.2 (0.0; 58.1)
Starch (g/day)	53.8 (8.0; 113.6)	59.7 (3.4; 129.8)
Sugar sweetened beverages (g/day)	37.5 (0.0; 275.0)	50.0 (0.0; 237.5)
DQI (units) ^1^	4.4 (3.5; 5.2)	4.4 (3.5; 5.3)
BMI Z-score (SD) ^2^	0.3 (−0.9; 1.7)	
Sleep (hours/day) ^2^	10.7	
Physical activity (% very active) ^2^	58.4	
Allocation (% intervention group) ^2^	44.8	
Maternal education (% medium education level) ^2,3^	52.8	
Maternal BMI ^2^	25.8 (20.5; 40.7)	
Maternal physical activity (% physically active at least 4 h per week) ^2^	59.6	

^1^: Diet quality index, ^2^: baseline measures of confounders, ^3^: Maternal education (“short-term further education [< 3 years]” or “medium-term further education [3–4 years]”).

**Table 2 ijerph-18-03590-t002:** Association between parental stress at baseline and subsequent change in child dietary intake and quality index.

Dietary Intake		Parental Stress	
	N	β (95% CI) *	P
Total energy (KJ/day)			
Crude	250	−1.70 (−56.45; 53.06)	0.95
Adjusted ^1^	250	−17.05 (−70.64; 36.55)	0.53
Protein (E%)			
Crude	250	−0.06 (−0.18; 0.05)	0.29
Adjusted ^1^	250	−0.05 (−0.17; 0.07)	0.41
Carbohydrates (E%)			
Crude	250	0.11 (−0.13; 0.35)	0.36
Adjusted ^1^	250	0.11 (−0.13; 0.35)	0.37
Added sugar (E%)			
Crude	250	0.05 (−0.13; 0.23)	0.57
Adjusted ^1^	250	0.07 (−0.11; 0.25)	0.43
Fat (E%)			
Crude	250	−0.05 (−0.27; 0.18)	0.67
Adjusted ^1^	250	−0.06 (−0.29; 0.17)	0.60
Saturated fat (E%)			
Crude	250	0.01 (−0.11; 0.13)	0.90
Adjusted ^1^	250	−0.01 (−0.13; 0.11)	0.84
Fruit (g/day)			
Crude	250	1.04 (−1.89; 3.98)	0.49
Adjusted ^1^	250	0.43 (−2.56; 3.43)	0.78
Vegetables (g/day)			
Crude	250	−1.63 (−4.85; 1.60)	0.32
Adjusted ^1^	250	−2.07 (−5.38; 1.25)	0.22
Fish (g/day)			
Crude	250	0.65 (−0.36; 1.65)	0.21
Adjusted ^1^	250	0.64 (−0.40; 1.67)	0.23
Starch (g/day)			
Crude	250	−0.86 (−2.62; 0.91)	0.34
Adjusted^1^	250	−1.16 (−2.97; 0.64)	0.20
Sugar sweetened beverages (g/day)			
Crude	250	1.24 (−3.41; 5.88)	0.60
Adjusted ^1^	250	0.99 (−3.75; 5.73)	0.68
DQI (units) ^§^			
Crude	250	0.01 (−0.2; 0.03)	0.53
Adjusted ^1^	250	0.00 (−0.02; 0.03)	0.75

* All analysis is adjusted for baseline measure of outcome. ^1^: Adjusted for (intervention/control), age, sex, BMI z-score, sleep, physical activity, and maternal BMI education, and physical activity. **^§^**: Diet quality index.

## Data Availability

In order to protect sensitive patient information, all data has been deposited in The Danish National Archives and is available upon online request through http://dda.dk/catalogue/22248 accessed on 4 February 2021. Archive number: 22248.

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
