# Peer review of "Associations between Parental Stress and Subsequent Changes in Dietary Intake and Quality among Preschool Children Susceptible to Obesity"

_ijerph, 2021, doi:10.3390/ijerph18073590_

Round 1
Reviewer 1 Report
A brief summary
The primary aim of this study was to examine the association between parental stress and changes in dietary intake among preschool children at risk for obesity.
Broad comments highlighting areas of strength and weakness.
Strengths
- An applicable topic given the prevalence of overweight and obesity and the role of parental figures in children’s eating patterns.
- Great use of person-first language (e.g., children with overweight).
- Utilized a picture book including to help participants obtain accurate food diary data.
Opportunities
Introduction
- Add citation for “Parents who are stressed may have a higher intake of unhealthy food and engage less in physical activity leading to increased sedentary behavior”
Materials and Methods
- Explain or cite why these criteria were chosen for “Eligibility criteria were being healthy weight but susceptible to future overweight defined as (i) having a high birth weight (>4000g) and/or (ii) having a mother who was overweight” prior to pregnancy (body mass index (BMI) ≥ 28 kg/m2).”
- Please provide additional information on the four-day estimated food diary used in this sentence “Children’s dietary intake was assessed using a four-day estimated food diary (Wednesday to Saturday) completed by the parents at baseline and after the 15 months of 116 follow-up.
Author Response
Thank you for the time spent reviewing our manuscript as well as for the constructive comments.
Add citation for “Parents who are stressed may have a higher intake of unhealthy food and engage less in physical activity leading to increased sedentary behavior”
Reply: We have now added a reference for this.
Explain or cite why these criteria were chosen for “Eligibility criteria were being healthy weight but susceptible to future overweight defined as (i) having a high birth weight (>4000g) and/or (ii) having a mother who was overweight” prior to pregnancy (body mass index (BMI) ≥ 28 kg/m2).”
Reply: Thank you for this comment. Previous research (Olsen et al and Danielzik et al, see ref below) has highlighted that children with a high birth weight, and/or having a mother who was overweight” prior to pregnancy are susceptible to future overweight. The aim of the healthy start intervention study was therefore to conduct a primary intervention to prevent future risk of obesity among normal weight children in risk of future overweight.
Olsen NJ, Mortensen EL, Heitmann BL. Predisposition to Obesity: Should We Target Those Most Susceptible? Curr Obes Rep. 2012 Mar;1(1):35-41. doi: 10.1007/s13679-011-0004-5. Epub 2012 Jan 25. PMID: 22448345; PMCID: PMC3302798.
Danielzik S, Czerwinski-Mast M, Langnase K, Dilba B, Muller MJ. Parental overweight, socioeconomic status and high birth weight are the major determinants of overweight and obesity in 5–7 y-old children: baseline data of the Kiel Obesity Prevention Study (KOPS) Int J Obes Relat Metab Disord. 2004;28:1494–1502. doi: 10.1038/sj.ijo.0802756
We have now quoted these articles at the end of the description of the inclusion criteria.
Please provide additional information on the four-day estimated food diary used in this sentence “Children’s dietary intake was assessed using a four-day estimated food diary (Wednesday to Saturday) completed by the parents at baseline and after the 15 months of 116 follow-up.
We have now added some more information on the food diaries:
“Children’s dietary intake was assessed using a four-day estimated food diary (Wednesday to Saturday) completed by the parents at baseline and after the 15 months of follow-up. These specific days were chosen to obtain information on dietary intake in both week- and weekend days. For each day the parents should register which food item was consumed, portion size, product brand and if possible, the fat content of the food item or meal. Furthermore, the parents should register the place and time for where and when the meal or food item was consumed. To help the parents estimate portion sizes, the food diaries were accompanied by a picture book including seventeen photo series with foods and portion sizes [21]. All information on dietary intake was manually entered, by one of the investigators, to the software program Dankost 3000 (http://dankost.dk) and nutritional calculations were performed to calculate the daily intake of macronutrients in grams and energy percent (E%). This software is based on the official Danish national food composition database (version 7.01) developed by the National Food Institute at the Technical University of Denmark [22]. Afterwards, means of total energy and macronutrient intake based on the four-day recorded food diaries was estimated for each child. Dietary intake of total energy, protein, fat, saturated fat, carbohydrate and added sugar was included in the analyses in units of KJ/day or as E%.
Because it was not possible to define food groups using the software Dankost 3000, this was done manually by studying a list of all food items eaten by the children. An algorithm was developed and programmed in Stata to be able to extract information and generate food groups (fruit, vegetables, sugar sweetened beverages (SSB), fish and starch). Food groups were defined based on standards developed by the National Food Institute, Technical University of Denmark [22] and adjusted to fit the Danish dietary guidelines (Supplementary table 2). Dietary intakes of fruit, vegetables, SSB, fish and starch were included in the analyses in units of g/day. Moreover, a diet quality index (DQI), adapted from Knudsen and colleagues [23], was calculated to examine to what degree the children complied with the Danish national dietary guidelines [24]. The DQI score was calculated based on six nutrients/food groups, as a function of recommended vs. reported intake, and included the following nutrients and food items: fat (maximum 30 E%), saturated fat (maximum 10 E%), added sugar (maximum 10 E%), fish (minimum 200 g/week), fruit and vegetables (minimum 300 g/day) and potatoes, rice or pasta (minimum 200 g/day). For the food groups with a minimum recommended intake, the group-score would be based on the ratio (R/RT) between the reported (R) and the recommended (RT) intake; where the score was set to 1 for intakes with R≥RT. For the food groups with a maximum recommended intake, the score was instead derived as 1-(R-RT)/(Rmax-RT); where the score is set to 1 for intakes R≤RT. The total DQI score ranged from 1 to 6, with 1 referring to values furthest away from following the recommendations, and with six referring to best compliance with the Danish national dietary recommendations. “
Reviewer 2 Report
Impacts of parental stress on children’s health behaviors and weight have been investigated and the relationships vary. In this longitudinal study, analytic data revealed a little association between parental stress and subsequent changes in dietary intake and quality of their children.
Risks of child obesity are multifactorial. Other concerns should be paid attention to the study.
- Preschool children susceptible to obesity are targeted population. What is the prevalence of child obesity in study area?
- Please explain the difference between intervention group and control group.
- High birth weight and maternal overweight represent stressful events to the parents. Maybe specific stressful events instead of general stress are of crucial role.
- Individual effect of maternal or paternal is of interests.
- Information of child number, parent education status, economic status is important to study outcomes.
Author Response
Thank you for the time spent reviewing our manuscript as well as for the constructive comments.
Risks of child obesity are multifactorial. Other concerns should be paid attention to the study.
1.Preschool children susceptible to obesity are targeted population. What is the prevalence of child obesity in study area?
Reply: Around 20% of the children from the Healthy Start population were overweight or obese at baseline.
2. Please explain the difference between intervention group and control group.
Reply: Families in the intervention group were offered the possibility to have up to ten sessions with a health consultant over a 15 month period focusing on improving dietary intake, physical activity and sleep and stress in the family. The intervention was tailor-made, meaning that the consultations were based on the family’s specific needs and resources. Families were also invited to participate in bi-monthly cooking and monthly play events, which took place at local schools and were located in five of the participating municipalities.
Families in the control group were invited to participate in a baseline examination and invited to follow-up examination approximate 15 month later and were not examined in between.
A description of this have now been added to the manuscript.
“The present study derived from the Danish Healthy Start primary intervention study [16] and the results presented stem from the prospective data collected over time. The “Healthy Start” investigated the joint effects of increasing physical activity, improving dietary habits, improve sleep habits and manage stress on child obesity prevention (ClinicalTrials.gov; NCT01583335). The intervention took place over on average period of 15 months and was conducted between 2009 and 2011. Families in the intervention group were offered the possibility to have up to ten sessions with a health consultant over a 15 month period focusing on stress, sleep, dietary intake and physical activity behaviour. The intervention was tailor-made, meaning that the consultations were based on the family’s specific needs and resources. Families were also invited to participate in bi-monthly cooking and monthly play events, which took place at local schools in five of the participating municipalities. Families in the control group were invited to participate in a baseline examination and invited to follow-up examination approximate 15 month later and were not examined in between.
Participants lived in 11 municipalities around the greater Copenhagen area and were between 2-6 years old at study entry. Eligibility criteria were being healthy weight but susceptible to future overweight defined as (i) having a high birth weight (>4000g) and/or (ii) having a mother who was overweight prior to pregnancy (body mass index (BMI) ≥ 28 kg/m2)[17]. A subgroup from one municipality was selected based on the criterion of having a mother with low educational level (≤ 10 years of education) [17, 18]. “
3.High birth weight and maternal overweight represent stressful events to the parents. Maybe specific stressful events instead of general stress are of crucial role.
Reply: We agree that high birth weight and maternal overweight may also introduce stress to the parents. Unfortunately, we do not have sufficient statistical power to stratify the analyses according to selection criteria. We also agree that specific stressful events may play a role in the investigated associations. However, specific stressful events are very difficult to target in an intervention, and therefore the Healthy Start intervention addressed (and measured) psychological stress experienced in everyday life.
A description of this have now been added to the manuscript.
“Furthermore, we did not have sufficient statistical power to stratify the analyses according to selection criteria or to investigate differences between those with extreme high scores of parental stress compared to those with extreme low parental stress scores.”
4. Individual effect of maternal or paternal is of interests.
Reply: While we agree that this is a good suggestion, given our limited sample size we simply don´t have the necessary statistical power to investigate individual maternal and paternal effects. We have added this as a limitation in the discussion section.
A description of this have now been added to the manuscript.
“Finally, we did not have the necessary statistical power to investigate individual maternal and paternal effects.”
5. Information of child number, parent education status, economic status is important to study outcomes.
Reply: Thank you, we have already adjusted our analysis for maternal education which can also be seen as a proxy for economic status.
Reviewer 3 Report
Thank you for the opportunity to review the paper. Here are my comments:
- It seems like there were over 60% of observations with missing data for relevant variables (250/635 had complete data). Did the authors consider multiple imputations as a sensitivity analysis to compare whether results differed between complete case analysis and multiple imputations?
- The statistical analyses section can be expanded to include information about descriptive analyses.
- Did the authors measure parental dietary behavior? That could be included as a covariate too.
- Are the authors able to tell which parent completed the stress questionnaire? I wonder if results would differ between whether it was the dad or mom
- Did the authors control for parental medical conditions such as mental health conditions, chronic conditions like diabetes, hypertension that would have an impact on stress level? Parental employment? Marital status?
Author Response
Thank you for the time spent reviewing our manuscript as well as for the constructive comments.
1. It seems like there were over 60% of observations with missing data for relevant variables (250/635 had complete data). Did the authors consider multiple imputations as a sensitivity analysis to compare whether results differed between complete case analysis and multiple imputations?
Reply: We have now conducted multiple imputation on missing data and it was possible to impute on 495 participants. The results are now presented as a sensitivity analyses (please see supplementary table 3).
“Sensitivity analyses: Especially at follow-up, we had a relatively large proportion of participants with missing information. Thus, in sensitivity analyses multiple imputations were used to impute missing values for all participants who had complete information on exposure and outcomes at baseline (n=495). The imputations were made using chained equations as implemented in Stata through the commands ice and mim [29]. For the multiple imputations, m =10 complete datasets were first generated. Baseline measure of dietary intake in addition to all preselected bassline covariates were used as imputation variables. In each dataset, the missing values was replaced with the imputed values, which were constructed based on predictive distributions for each of the missing values. Afterwards each of the completed datasets was analysed, and the results from the ten analyses were combined to create a single set of estimates comprising the variability associated with the missing values. As in the primary analyses, linear regressions were used to explore both crude and adjusted associations.”
2. The statistical analyses section can be expanded to include information about descriptive analyses.
Reply: We have now added information on this.
“Descriptive analyses were carried out on baseline and follow-up measures of exposure and outcome and on baseline measures of confounders.”
3.Did the authors measure parental dietary behavior? That could be included as a covariate too.
Reply: Information on parental dietary behaviour was not obtained.
4. Are the authors able to tell which parent completed the stress questionnaire? I wonder if results would differ between whether it was the dad or mom
Reply: No, we do not have information on which parent that completed the stress questionnaire. This is also discussed under strength and limitations to the study.
“Furthermore, we had no information on which parent (mother or father) completed the questions on perceived stress. However, we suspect that the ranking of stress level among mothers and fathers to be the same and by that not affecting our results.”
5. Did the authors control for parental medical conditions such as mental health conditions, chronic conditions like diabetes, hypertension that would have an impact on stress level? Parental employment? Marital status?
Reply: We do not have information on parental medical conditions such as mental health conditions, chronic conditions like diabetes, hypertension. We have information on maternal employment status, however, we have decided to adjust for maternal education, which can be seen as a proxy for maternal employment status.
Reviewer 4 Report
Reviewer's report
This manuscript addresses an interesting topic in understanding the association between parental stress and dietary intake among preschool children. However, the paper is not suitable for publication in its current form. The authors will need to address some major points and have the option to submit a revised version. The matters to be addressed can be found below.
Major Comments:
Introduction:
The authors need to make introduction easier and clearer to help the reader follow the study hypothesis and the gaps in the literature that this study aims to fill in.
- For lines 50-56 please add references to support the statements. In the second paragraph they report available studies, mainly cross-sectional.
- Lines 58- 61: the authors report the findings by Bauer et al. but they need to clarify ‘‘how the decreased time to attend to their own nutrition’’ is related to children’s diet.
- Could you please elaborate more how hair cortisol is related to parental stress?
- Line 66: ‘‘Despite evidence for an association…’’ The previously mentioned study with Ref 13 showed no association. Please clarify.
- Line 70: it is not clear which are the gaps. Is it only the lack of longitudinal studies?
- Lines 70-72: The authors state that the aim of the study is to examine ‘‘the association between parental stress and subsequent changes in dietary intake and quality among preschool children susceptible to obesity’’. What do they mean by children susceptible to obesity? Is there any rationale to support the selection of this specific group? Nothing has been mentioned up to this point for susceptibility to obesity.
Methodology:
- Lines 81-83: Is there a reference to support the criteria for susceptibility to obesity?
- Line 84: Do the authors mean low education level?
- Line 95: The authors report that the final population is 250 children. How many children from the intervention and control group?
- Lines 106-107: What do the authors mean by ‘‘inspiration for these questions?’’ They selected specific questions? If yes, based on what criteria?
- Lines 185-185: Please define the two categories.
Results:
- Table 2: did the authors adjust for BMI or BMI z-scores?
Discussion:
- Lines 274-276: The authors mention ‘‘… or as we did, a modified version or a subscale focusing of a specific domains for example financial strain, mental health or stress within the family.’’ and ref 24. In the measures section the authors mention ‘‘inspiration for these questions was obtained from the Swedish version of the Parenting Stress Index (PSI)’’ and ref 11. Can you please clarify the parental stress assessment?
- Even though the authors adjusted for intervention/ control, observations may be affected if the majority of children come for the intervention group.
Author Response
Thank you for the time spent reviewing our manuscript as well as for the constructive comments.
1. For lines 50-56 please add references to support the statements. In the second paragraph they report available studies, mainly cross-sectional.
Reply: We have ow added references to support the statements.
2. Lines 58- 61: the authors report the findings by Bauer et al. but they need to clarify ‘‘how the decreased time to attend to their own nutrition’’ is related to children’s diet.
Reply: We have re-written the sentence.
“This was further supported by a cross-sectional study from Bauer et al. that reported that parental employment and workload contributed to decreased time to prepared meals and less frequent family meals as well as more unhealthy food choices [13].”
3. Could you please elaborate more how hair cortisol is related to parental stress?
Reply: Cortisol measured in blood, saliva or urine has been the most used biological measures of stress however, multiple samples taken during 24 hours for several days are needed for a reliable measure of stress level. Hair cortisol concentration is a relatively new biomarker, providing a biological measure of long-term cortisol exposure as cortisol is embedded into hair as it grows. Some studies have suggested a direct relationship between hair cortisol concentration and adiposity among children and adults.
Dauegaard S, Olsen NJ, Heitmann BL, Larsen SC. Familial associations in hair cortisol concentration: A cross-sectional analysis based on the Healthy Start study. Psychoneuroendocrinology. 2020 Nov;121:104836. doi: 10.1016/j.psyneuen.2020.104836. Epub 2020 Aug 19. PMID: 32858307.
Larsen SC, Fahrenkrug J, Olsen NJ, Heitmann BL. Association between Hair Cortisol Concentration and Adiposity Measures among Children and Parents from the "Healthy Start" Study. PLoS One. 2016 Sep 23;11(9):e0163639. doi: 10.1371/journal.pone.0163639. PMID: 27662656; PMCID: PMC5035005.
Maria Elena Brianda, Isabelle Roskam, Moïra Mikolajczak, Hair cortisol concentration as a biomarker of parental burnout, Psychoneuroendocrinology, Volume 117,2020
We have added a describtion of this in the discussion.
“Hair cortisol concentration is a relatively new biomarker [34, 35], providing a biological measure of long-term cortisol exposure as cortisol is embedded into hair as it grows and when comparing to objective measures of parental stress, the results from this current study are in accordance with our results investigating the cross-sectional association between parental hair cortisol levels and intake of total energy, added sugar, selected food groups or DQI among their children, where we also could not demonstrate and association among 296 children [15].”
4. Line 66: ‘‘Despite evidence for an association…’’ The previously mentioned study with Ref 13 showed no association. Please clarify.
Reply: This sentence is related to the next section and not related to the study results from ref 13.
We have re-written the sentence. Please see manuscript or below.
“Despite some evidence from cross-sectional studies showing an association between parental stress and child dietary intake, knowledge from longitudinal studies on the association between these underlying factors contributing to the obesity epidemic is sparse among preschool children.“
5. Line 70: it is not clear which are the gaps. Is it only the lack of longitudinal studies?
Reply: The primary gap is the lack of longitudinal studies which is necessary to show the casual relationship between parental stress and child dietary behaviour.
“Despite some evidence from cross-sectional studies showing an association between parental stress and child dietary intake, knowledge from longitudinal studies on the association between these underlying factors contributing to the obesity epidemic is sparse among preschool children.“
6. Lines 70-72: The authors state that the aim of the study is to examine ‘‘the association between parental stress and subsequent changes in dietary intake and quality among preschool children susceptible to obesity’’. What do they mean by children susceptible to obesity? Is there any rationale to support the selection of this specific group? Nothing has been mentioned up to this point for susceptibility to obesity.
Reply: Thank you for your comment. Children who are susceptible to obesity are children who are predisposed to obesity, in this case children who had a high birth weight, came from low SES families or had a mother who were overweight prior to pregnancy. These references have now been added to the manuscript.
“To address these gaps, the current study used longitudinal data from the “Healthy Start” study [16] to examine the association between parental stress and subsequent changes in dietary intake and quality among preschool children predisposed to obesity.”
Olsen NJ, Mortensen EL, Heitmann BL. Predisposition to Obesity: Should We Target Those Most Susceptible? Curr Obes Rep. 2012 Mar;1(1):35-41. doi: 10.1007/s13679-011-0004-5. Epub 2012 Jan 25. PMID: 22448345; PMCID: PMC3302798.
Danielzik S, Czerwinski-Mast M, Langnase K, Dilba B, Muller MJ. Parental overweight, socioeconomic status and high birth weight are the major determinants of overweight and obesity in 5–7 y-old children: baseline data of the Kiel Obesity Prevention Study (KOPS) Int J Obes Relat Metab Disord. 2004;28:1494–1502. doi: 10.1038/sj.ijo.0802756
Methodology:
1. Lines 81-83: Is there a reference to support the criteria for susceptibility to obesity?
Reply: Previous research (Olsen et al and Danielzik et al, see ref below) has highlighter that children with a high birth weight, and/or having a mother who was overweight” prior to pregnancy are susceptible to future overweight. The aim of the healthy start intervention study was therefore to conduct a primary intervention to prevent future risk of obesity among normal weight children.
Olsen NJ, Mortensen EL, Heitmann BL. Predisposition to Obesity: Should We Target Those Most Susceptible? Curr Obes Rep. 2012 Mar;1(1):35-41. doi: 10.1007/s13679-011-0004-5. Epub 2012 Jan 25. PMID: 22448345; PMCID: PMC3302798.
Danielzik S, Czerwinski-Mast M, Langnase K, Dilba B, Muller MJ. Parental overweight, socioeconomic status and high birth weight are the major determinants of overweight and obesity in 5–7 y-old children: baseline data of the Kiel Obesity Prevention Study (KOPS) Int J Obes Relat Metab Disord. 2004;28:1494–1502. doi: 10.1038/sj.ijo.0802756
We have now quoted these articles at the end of the description of the inclusion criteria’s
2. Line 84: Do the authors mean low education level?
Reply: We have corrected this.
“A subgroup from one municipality was selected based on the criterion of having a mother with low educational level (≤ 10 years of education). “
3. Line 95: The authors report that the final population is 250 children. How many children from the intervention and control group?
Reply: In total 112 children participated from the intervention group and 138 from the control group. This information has now been added.
“…therefore the final study population for analysis consisted of 250 (n=112 intervention group; n=138 control group) children aged 2-6 years”
4. Lines 106-107: What do the authors mean by ‘‘inspiration for these questions?’’ They selected specific questions? If yes, based on what criteria?
Reply: We have tried to clarify this and re-written.
“Information on parental stress was obtained through a questionnaire completed by one of the parents at baseline of the “Healthy Start” study [16]. The Parental Stress Index, which is a self-reported inventory, designed to measure parental experiences of stress in relation to the parent–child interaction was used to assess parental stress. In the Healthy Start project, a modified Swedish version [19] was included, as an elaboration of the parental part of the questionnaire. In total 10 questions were selected from the Swedish version and modified according to context (Supplementary table 1).
The questions concerned which changes the parents had experienced in their daily living after they had had the child, in relation to sleep, stress, worries, time for themselves, household conflicts, workload, social gatherings in the home, joy of life, energy in daily living, and perceived complexity of being a parent compared to expectations.
Answers were coded 0-2 with 0 being the best score and 2 the worst score e.g. less stress=0, more stress=2 or no change in stress level=1. Analyses of intercorrelations and factor analyses showed that nine (workload excluded) out of the ten questions could be added together to an overall score measure of parental stress [20], with a higher score indicating higher levels of perceived stress among the parents.”
5. Lines 185-185: Please define the two categories.
Reply: We have added a further description of this.
“The parents could answer if they 1) conducted vigorous physical activity several times per week, 2) did moderate physical active at least 4 hours per week (exercise, bike to work, heavy gardening), 3) light physical activity at least 4 hours per week (walking, light gardening) or 4) primarily sedentary activity. For this study, information was collapsed into two categories “trained hard serval times per week” (answer 1 and 2) and “being physical active for at least 4 times per week” (answer 3 and 4).“
Results:
1. Table 2: did the authors adjust for BMI or BMI z-scores?
Reply: As described in the statistical section, we adjusted for BMI Z-score
Discussion:
1. Lines 274-276: The authors mention ‘‘… or as we did, a modified version or a subscale focusing of a specific domains for example financial strain, mental health or stress within the family.’’ and ref 24. In the measures section the authors mention ‘‘inspiration for these questions was obtained from the Swedish version of the Parenting Stress Index (PSI)’’ and ref 11. Can you please clarify the parental stress assessment?
Reply: Thank you. We have tried to clarify the parental stress assessment.
“Information on parental stress was obtained through a questionnaire completed by one of the parents at baseline of the “Healthy Start” study [16]. The Parental Stress Index, which is a self-reported inventory, designed to measure parental experiences of stress in relation to the parent–child interaction was used to assess parental stress. In the Healthy Start project, a modified Swedish version [19] was included, as an elaboration of the parental part of the questionnaire. In total 10 questions were selected from the Swedish version and modified according to context (Supplementary table 1).
The questions concerned which changes the parents had experienced in their daily living after they had had the child, in relation to sleep, stress, worries, time for themselves, household conflicts, workload, social gatherings in the home, joy of life, energy in daily living, and perceived complexity of being a parent compared to expectations.
Answers were coded 0-2 with 0 being the best score and 2 the worst score e.g. less stress=0, more stress=2 or no change in stress level=1. Analyses of intercorrelations and factor analyses showed that nine (work-load excluded) out of the ten questions could be added together to an overall score measure of parental stress [20], with a higher score indicating higher levels of perceived stress among the parents.”
2. Even though the authors adjusted for intervention/ control, observations may be affected if the majority of children come for the intervention group.
Reply: There were more children participating from the control group compared to the intervention group (n=112 intervention group; n=138 control group). As described in the statistical section, potential effect modification by group allocation (intervention/control group) was explored for all outcomes by adding a product term (baseline outcome*group allocation) to the models. We found no such interaction, which is also presented in the result section.
“We further found no evidence of interaction between group (intervention/control) or child gender and parental stress in relation to intake of energy, macronutrients, food groups or DQI.”
Reviewer 5 Report
Review report: Associations between parental stress and subsequent changes in dietary intake and quality among preschool children susceptible to obesity
The paper examines the association between parental stress and changes in dietary intake and quality among preschool children susceptible to obesity. With the data from the Healthy Start study, which cover 250 preschool children, the study unveils the association between parental stress and changes in dietary intake and quality for children with the adoption of multiple linear regressions.
Generally, this paper is interesting, and the structure is also clear. However, there are some minor revisions are needed for the publication consideration.
- The data processing in present paper, especially the dietary data, is not clear enough. more information on the survey and data processing are necessary.
- The paper performs a longitudinal study, to my understanding, dataset would be panel data. Some panel data skills would be employed to carry out the empirical results. The authors need to improve the analysis techniques.
- It is highly recommended to tear up the data and perform statistic tests to check the differences between the groups with high and low Parental stress. This is more interesting to readers. From my point of view, the diet habit would be quite consistent in short time. Thus, the impacts of parental stress would have insignificant effect on the dietary intake in short run. The discussion part is necessary to be improved.
- Proofreading needs to be done before publication
Author Response
Thank you for the time spent reviewing our manuscript as well as for the constructive comments.
1. The data processing in present paper, especially the dietary data, is not clear enough. more information on the survey and data processing are necessary.
Reply: We have now added some further information on the processing of dietary data.
“Children’s dietary intake was assessed using a four-day estimated food diary (Wednesday to Saturday) completed by the parents at baseline and after the 15 months of follow-up. These specific days were chosen to obtain information on dietary intake in both week- and weekend days. For each day the parents should register which food item was consumed, portion size, product brand and if possible, the fat content of the food item or meal. Furthermore, the parents should register the place and time for where and when the meal or food item was consumed. To help the parents estimate portion sizes, the food diaries were accompanied by a picture book including seventeen photo series with foods and portion sizes [21]. All information on dietary intake was manually entered, by one of the investigators, to the software program Dankost 3000 (http://dankost.dk) and nutritional calculations were performed to calculate the daily intake of macronutrients in grams and energy percent (E%). This software is based on the official Danish national food composition database (version 7.01) developed by the National Food Institute at the Technical University of Denmark [22]. Afterwards, means of total energy and macronutrient intake based on the four-day recorded food diaries was estimated for each child. Dietary intake of total energy, protein, fat, saturated fat, carbohydrate and added sugar was included in the analyses in units of KJ/day or as E%.
Because it was not possible to define food groups using the software Dankost 3000, this was done manually by studying a list of all food items eaten by the children. An algorithm was developed and programmed in Stata to be able to extract information and generate food groups (fruit, vegetables, sugar sweetened beverages (SSB), fish and starch). Food groups were defined based on standards developed by the National Food Institute, Technical University of Denmark [22] and adjusted to fit the Danish dietary guidelines (Supplementary table 2). Dietary intakes of fruit, vegetables, SSB, fish and starch were included in the analyses in units of g/day. Moreover, a diet quality index (DQI), adapted from Knudsen and colleagues [23], was calculated to examine to what degree the children complied with the Danish national dietary guidelines [24]. The DQI score was calculated based on six nutrients/food groups, as a function of recommended vs. reported intake, and included the following nutrients and food items: fat (maximum 30 E%), saturated fat (maximum 10 E%), added sugar (maximum 10 E%), fish (minimum 200 g/week), fruit and vegetables (minimum 300 g/day) and potatoes, rice or pasta (minimum 200 g/day). For the food groups with a minimum recommended intake, the group-score would be based on the ratio (R/RT) between the reported (R) and the recommended (RT) intake; where the score was set to 1 for intakes with R≥RT. For the food groups with a maximum recommended intake, the score was instead derived as 1-(R-RT)/(Rmax-RT); where the score is set to 1 for intakes R≤RT. The total DQI score ranged from 1 to 6, with 1 referring to values furthest away from following the recommendations, and with six referring to best compliance with the Danish national dietary recommendations. “
2. The paper performs a longitudinal study, to my understanding, dataset would be panel data. Some panel data skills would be employed to carry out the empirical results. The authors need to improve the analysis techniques.
Reply: The most appropriate statistical method will depend on the underlying hypotheses. We have used a multivariate linear regression model to explore baseline parental stress as a predictor of subsequent change in their children’s dietary intake. In our opinion there are no other statistical methods which would be more appropriate for this purpose.
3. It is highly recommended to tear up the data and perform statistic tests to check the differences between the groups with high and low Parental stress. This is more interesting to readers. From my point of view, the diet habit would be quite consistent in short time. Thus, the impacts of parental stress would have insignificant effect on the dietary intake in short run. The discussion part is necessary to be improved.
Reply: While we agree that this is a good suggestion, given our limited sample size we simply don´t have the necessary statistical power to focus specifically on participants at the extremes of the parental stress scale (e.g. high and low specifically). We have added this as a limitation in the discussion section.
“Furthermore, we did not have sufficient statistical power to stratify the analyses according to selection criteria or to investigate differences between those with extreme high scores of parental stress compared to those with extreme low parental stress scores.”
4. Proofreading needs to be done before publication
Reply: We have done our upmost to correct all typos and grammatical errors.
Round 2
Reviewer 2 Report
There is no additional comment.
Reviewer 4 Report
I have no further comments.